# Empirical Analysis on Public Expenditure for Education, Human Capital and Economic Growth: Evidence from Honduras

**Roldán Villela *** and **Juan Jacobo Paredes**

Faculty of Graduate Studies, Universidad Tecnologica Centroamericana (UNITEC), Tegucigalpa 11101, Honduras
* Correspondence: roldan.villela@unitec.edu

**Abstract:** The objective of this study is to evaluate the relationship between public expenditure for education and human capital on economic growth in Honduras from 1990 to 2020, using the instrumental variables (IV) method, which incorporates the components of public spending on education and human capital, in addition to a set of control variables. The time series were extracted from the World Bank online databases. The results show that there is no correlation between public expenditure for education and economic growth; they also suggest that human capital is not contributing to economic growth, confirming that human capital accumulation is not fully developing. Finally, of the set of control variables considered key by the literature and on which social and economic development depends to a large extent, these would be preventing sustained economic growth, so the government and the population have enormous challenges to overcome.

**Keywords:** public expenditure; education; human capital; economic growth





## 1. Introduction

One of the main determinants of a country's economic success is education (Barro 1991), which is why investment in education is probably the instrument most used by states to achieve a country's economic growth. The investment in education prepares its citizens to obtain a comprehensive education; it also fosters and promotes science, research and technology. In this way, it is possible to improve people's standard of living (Zárate and Gómez 2011).

For his part, Weisbrod (1962) stated that human capital plays a fundamental role in achieving sustained economic growth. Recognizing the responsibility of the public sector in this task, he also affirmed that investment in future productivity is increasingly produced outside the private market and in intangible forms. Similarly, Schultz (1961) stated that the most relevant factors for the improvement of the social welfare of the population are advances in knowledge and the continuous improvement of competencies and skills, above capital and land, and that spending on education aimed at improving the skills of workers will increase the value of productivity, generating positive returns.

Honduras is experiencing multiple crises in the educational, social, economic, political and environmental spheres, which are manifested and reinforced simultaneously, including insufficient economic growth with little impact on reducing inequalities and poverty and high levels of informality and imbalance in public finances (PNUD 2021). The rule of law has been systematically weakened over the last 30 years, with recurrent governance crises that have affected the population, damaged the state's capacity to respond to growing social demands and slowed progress in human capital. In 2020, the human capital index of Honduras, according to the World Bank, was 0.48, one of the lowest in Latin America, only surpassed by Haiti and Guatemala. For its part, the results of the PISA D tests conducted in 2018 in Latin American countries indicate that Honduras ranks second to last in the region, only ahead of the Dominican Republic (OCDE 2018), thus evidencing an acute learning

crisis, if one considers that Latin America is in turn one of the most lagging regions in the world in terms of educational performance. Much has been said about the country's corrupt practices. The educational sector is no exception. Efforts to improve the efficiency of education spending will undoubtedly contribute to significant improvements in human capital skills, competencies and productivity.

The Honduran government invests in education at a level similar to other countries in the region. Despite this, it has one of the lowest per capita incomes and human capital indices on the continent today. At constant prices, GDP per capita in Latin America and the Caribbean maintained an upward trend. Between 1990 and 2020, there was a significant increase of 38%, that is, from USD 5801.60 to USD 8000.20, respectively. Particularly, the per capita GDP of Honduras for the same period had an increase that could be considered significant of 29%, from USD 1738.60 in 1990 to USD 2239 in 2020 (World Bank 2022). Despite this, it continues in the category of a low-middle-income country and very far from some countries of the isthmus. Costa Rica went from USD 6041.40 in 1990 to USD 12,126.60 in 2020, evidencing an increase of USD 6085.20, equivalent to an extraordinary increase of more than 100%, evidently much higher than that recorded by Honduras. Similar is the case of Panama, whose per capita income in 1990 was USD 5142.20; in 2020, it reached USD 12,172.30, which translates into an impressive increase that also exceeds 100%, obtaining a remarkable performance at the regional and world level, placing both nations, according to the World Bank's ranking of per capita income, in upper-middle-income countries.

Analyzing public spending on education in Honduras expressed both as a percentage of total government spending and of GDP in the research period reflects a generally stationary trend, despite experiencing a sustained economic growth rate in that period (except for the already known crises). However, there was a slight increase in the education budget allocated by the state. In the period 1990–2020, public spending on education as a percentage of total government spending increased from 19.7% to 22.75%. With respect to GDP, it went from 3.8% in 1990 to 6.37% in 2020, with a significant increase starting in 2001.

This study aims to place the importance of public spending on education on the government agenda, exposing the potential it provides to the state and its own citizens in increasing production through investment in human capital. The results of this study provide empirical evidence to decision makers, donors and academia to generate and implement public policies aimed at increasing economic growth through education. Against this background, the objective of this study is to evaluate the relationship between public spending on education and human capital on economic growth in Honduras between 1990 and 2020, using an econometric model that incorporates public spending on education and human capital in economic growth models.

## 2. Literature Review

According to the Honduran Ministry of Finance, public spending on education is aimed at developing intellectual, physical and moral skills through the provision of educational services to students at all levels, including preschool, primary, secondary, higher education and vocational training, as well as the provision of cultural, sports and research services. One of the reasons that justify state intervention to ensure equal opportunities for the entire population is the divergence between the private and social rate of return on education.

Ziberi et al. (2022), using instrumental variables, found a positive relationship between public spending on education and economic growth in North Macedonia. The research considered GDP as the dependent variable. The research period was from 1917 to 2020. In the same study conducted by Le and Tran (2021) in Vietnam for the period 2006 to 2019, the research results show that there is a bidirectional nexus between economic growth and public expenditure on education with a lag of about two years. For this, they used the vector autoregression (VAR) model and the Granger causal model. Okerekeoti (2022) examined the effect of public expenditure on education on economic growth in Nigeria from 1999 to 2020. Using regression analysis concluded that there is a positive and significant effect

between public expenditure on education and real GDP. Osiobe (2020) used the panel unit root test and panel cointegration analysis for the period 2000–2014, which includes eight Latin American countries. The results concluded that public spending on education and economic growth in the selected countries are positively and significantly associated, in the long and short run.

Evaluating a 30-year period, from 1990 to 2020, Nuţă et al. (2022), analyzed the impact of public spending on education on economic growth in 11 former communist states of Eastern Europe. The methodology used was autoregressive distributed lag (ARDL) with structural breaks. The results show that the relationship between public spending on education and economic growth is mixed in the long run. For five countries, there was no relationship. In contrast, for six countries, there was a relationship in the long run. In the short term, there were also mixed results; for four countries, the relationship was positive, and for two, the relationship was negative. For their part, Gheraia et al. (2021), in their study, revealed that spending on education in the Kingdom of Saudi Arabia had a positive effect on economic growth for the period 1990–2017. Meanwhile, Govindaraju et al. (2011) stated that public spending on education is an important determinant of GDP growth. The education sector will increase GDP growth by improving productivity. They examined time series data for Malaysia over the period 1970 to 2006. They applied bivariate and multivariate models to study the relationship between public expenditure on education and GDP.

Among some studies that consider human capital and public spending on education are Baldacci et al. (2004), who used a recursive system of equations to examine the direct and indirect channels linking public spending on education, human capital and economic growth, using a sample of 120 developing countries between 1975 and 2000. The results show that public spending on education has a positive and significant direct impact on human capital accumulation and, consequently, on higher economic growth. Dissou et al. (2016) developed a multisectoral endogenous growth model with human capital accumulation in addition to other fiscal instruments; they evaluated the implications for growth of alternative methods of financing public spending on education in Benin. Their results found significant differences in the choice of financing method; the non-distortionary financing method provides the largest increase in output thanks to its strong effect on physical and human capital stocks.

Shafuda and De (2020) examined the impact of public spending on human capital on human development indicators such as health care outcomes, educational attainment and national income growth in Namibia using time series data from 1980 to 2015. The results were mixed, revealing a significant positive long-run relationship of public spending on education with primary net enrollment rate and tertiary gross enrollment rate. In contrast, no cointegration was observed between public education spending and primary and secondary gross enrollment rate. Using a panel methodology, Pelinescu (2015) tried to show the role of human capital as a growth factor., The model revealed a positive, statistically significant relationship between GDP per capita and the innovative capacity of human capital, evidenced by the number of patents and the qualification of employees.

Moreover, Linhartová (2020) conducted a study with the aim of verifying whether public investment in areas that develop human capital can effectively help its development in the Czech Republic during the period from 1995 to 2018. They used the method of least squares; the findings showed that spending on education and health ranked only third or fourth in terms of their contribution to the development of human capital. Using the error correction model as an analytical tool, Kanayo (2013) empirically examined the relationship between economic growth and human capital development in Nigeria. The results show that investment in human capital in the form of education and skill development at the primary and secondary levels has a significant impact on economic growth. Guarnizo (2018) analyzed the relationship between human capital (measured by the schooling rate) and economic growth (measured by GDP) in Colombia, during the period 1980–2015, using cointegration techniques, the autoregressive vector model and the error correction vector

model, with which she obtained, as a result, a short- and long-term relationship between both variables. She found no causal effects in either direction. She concludes, as does (Romer 1986), that the accumulation of human capital undoubtedly increases the levels of economic growth, which in turn improves technology and innovation.

The review of these studies tends to show a clear trend in favor of the positive effects of public spending on education and human capital on economic growth, as well as on the generation of beneficial effects for individuals and society as a whole. On the other hand, it is necessary to emphasize, as the literature suggests, that it is ideal to analyze the efficiency of spending and governance in order to have a more objective assessment.

## 3. Methodology

To understand the effect of public expenditure on education on economic growth, a model similar to that used by several authors (Ram 1986; Lucas 1988; Barro 1990; Barro and Sala-I-Martin 1992), who incorporated public expenditure in an endogenous growth model, is implemented. An equation is used to identify the correlations between expenditure on education, human capital and economic growth. This scheme provides concrete information on how public expenditure on education and human capital are related to economic growth.

Given the relationships found by Lucas between education and economic growth, and between educational level and education expenditure, it is necessary to implement the instrumental variables (IV) method as a measure to solve the endogeneity problem. With this tool, conflicts due to biased or inconsistent parameter estimation are resolved through the two-stage OLS technique (Wooldridge 2009). A variable is endogenous if it is correlated with the disturbance term; the presence of an endogenous variable among the regressors of a model requires the use of instruments. For the estimator to become consistent, an instrument is used that is correlated with the endogenous variable but not with the disturbance term. As used by Urhie (2014), the urban population regressor is used as an instrument. This is linked to the secondary gross enrollment rate (endogenous) because the accelerated population increase in cities will overwhelm the existing educational infrastructure, preventing the entry of more students.

The model used in this study is configured as follows:

$$y = f(h, g, c) \tag{1}$$

where:

y = GDP per capita growth rate (GDPPC),
h = variables related to human capital,
g = public expenditure for education measures,
c = control variables.

Econometrically, the equation is expressed as follows:

$$\gamma = \beta o + \alpha_1 \|_1 + \beta_2 \}_2 + \delta_3 \rfloor_3 + \sqcap \tag{2}$$

As for the set of regressors related to human capital (h), these are consistent with those contained in the methodology developed by the (World Bank 2020) to estimate the human capital index. This methodology involves three components (survival, schooling and health). For the survival component, the under-five mortality rate was used. For the schooling component, gross enrollment rates at all educational levels were used as a proxy, as used by (Barro 1991). For the health component, the prevalence of stunting, height for age and survival at age 65, in men and women, were used. An important consideration is to mention that the method created by the World Bank to calculate the human capital index is recent, created in 2017. Therefore, it is not possible to use this index for previous years, nor is it possible to construct the index due to the absence of information on several of the components of the methodology in Honduras. However, studies such as that of (Barro 2001) use several of the variables contained in the methodology to estimate the human capital index and in the present research.

The set of variables (g) is made up of a ratio of government expenditure on education to total government spending and a ratio of government expenditure on education to GDP. Equal variables were incorporated by other authors (Urhie 2014; Marmullaku et al. 2020).

The set of regressors (c) is constituted by variables that are usually incorporated in economic growth models. The ones used in this study are GDP per capita, GDP growth, inflation, unemployment, gross fixed capital formation, trade, financial depth and inequality.

The instrument used was the urban population (URBAN), after making multiple attempts with life expectancy and the rate of dependents, which are the most common instruments identified for this purpose; however, multicollinearity problems were encountered, and combinations of the 3 variables were tried without obtaining satisfactory results. The model used is based on the endogenous growth theory of (Lucas 1988), as well as on several empirical research results associated with the study of education spending, human capital and economic growth. The instrumental variables method has been widely used in these types of studies (Barro 1999; Barro 2001; Urhie 2014; Marmullaku et al. 2020; Ziberi et al. 2022).

The regression error u is thought to be unrelated to g and c but related to h. This correlation arises from the simultaneity bias caused by the concurrent relationship between economic growth and education, as illustrated by the Lucas model. Because of this correlation, the ordinary least squares (OLS) estimator is biased and inconsistent for $\beta$. To obtain a consistent estimator, we assume the existence of at least one instrumental variable z that satisfies the assumption

$$E\,(u/z) = 0$$

This is a requirement for instrument validity. Furthermore, the instrument z must be correlated with h in order to provide information on the variables being instrumented. Aside from the concurrent relationship between education and economic growth, two other factors—omitted-variable bias and variable errors—could lead to a violation of the zero-conditional mean assumption in economic research. Despite the fact that each of these issues arises for different reasons, the solution is the same econometric tool: the instrumental variables (IV) estimator.

Endogenous variables are those that are correlated with the disturbance term. When an endogenous variable is present among the regressors in a model, instrumental variables or instruments must be used. This has been shown to solve the problem of biased and inconsistent parameter estimates caused by the use of the OLS technique (Wooldridge 2009). Table 1 below shows the variables used in this research.

**Table 1.** Variables used.

| No. | Variables | Name of the Variables | Definitions of Variables | Source |
|---|---|---|---|---|
| 1 | GDPPC | GDP per capita growth (annual %) | Annual percentage growth rate of GDP per capita based on constant local currency. | WB |
| 2 | RGDPPC | GDP per capita (constant USD) | GDP per capita is gross domestic product divided by midyear population. | WB |
| 3 | GDP | GDP growth (annual %) | Annual percentage growth rate of GDP at market prices based on constant local currency. | WB |
| 4 | INFLA | Inflation (annual %) | Measured by the consumer price index; reflects the annual percentage change in the cost to the average consumer of acquiring a basket of goods and services. | WB |
| 5 | UNEMP | Unemployment (% of total labor force) | Unemployment refers to the share of the labor force that is without work but available for and seeking employment. | WB |

**Table 1.** *Cont.*

| No. | Variables | Name of the Variables | Definitions of Variables | Source |
|---|---|---|---|---|
| 6 | GFCF | Gross fixed capital formation (% of GDP) | Includes land improvements; plant, machinery and equipment purchases; and the construction of roads, railways, schools, offices, hospitals, private residential dwellings and buildings. | WB |
| 7 | TRADE | Trade (% of GDP) | Trade is the sum of exports and imports of goods and services measured as a share of gross domestic product. | WB |
| 8 | FDEPTH | Financial depth (ratio of broad money % of GDP) | Sum of currency outside banks; demand deposits other than those of the central government; the time, savings and foreign currency deposits of resident sectors other than the central government. | WB |
| 9 | INEQUA | Inequality (Gini index) | Gini index measures the extent to which the distribution of income among individuals or households within an economy deviates from a perfectly equal distribution. | WB |
| 10 | EDUGDP | Government expenditure on education (% of GDP) | General government expenditure on education (current, capital and transfers) is expressed as a percentage of GDP. It includes expenditure funded by transfers from international sources to government. | WB |
| 11 | EDUEXP | Government expenditure on education (% of government expenditure) | General government expenditure on education (current, capital and transfers) is expressed as a percentage of total general government expenditure on all sectors (including health, education, social services, etc.). | WB |
| 12 | SCHPRE | School enrollment, preprimary (% gross) | Ratio of total enrollment, regardless of age, of the population of the age group that officially corresponds to the level of education shown. Total enrollment in preprimary education. | WB |
| 13 | SCHPRI | School enrollment, primary (% gross) | Total enrollment in primary education. | WB |
| 14 | SCHSEC | School enrollment, secondary (% gross) | Total enrollment in secondary education. | WB |
| 15 | SCHTER | School enrollment, tertiary (% gross) | Total enrollment in tertiary education. | WB |
| 16 | STUNT | Prevalence of stunting, height for age (% of children under 5) | Prevalence of stunting is the percentage of children under age 5 whose height for age is more than two standard deviations below the median for the international reference population ages 0–59 months. | WB |
| 17 | MORTAL | Mortality rate, under 5 (per 1000 live births) | Under-five mortality rate is the probability per 1000 that a newborn baby will die before reaching age five, if subject to age-specific mortality rates of the specified year. | WB |
| 18 | SURVIF | Survival to age 65, female (% of cohort) | Survival to age 65 refers to the percentage of a cohort of newborn infants that would survive to age 65, if subject to age specific mortality rates of the specified year. | WB |
| 19 | SURVIM | Survival to age 65, male (% of cohort) | IDEM | WB |
| 20 | URBAN | Urban population (% of total population) | Urban population refers to people living in urban areas as defined by national statistical offices. | WB |

The definitions, information and data for the variables evaluated in this study were extracted from the World Bank's online databases.

## 4. Results

Table 2 in this section presents the results obtained after running the econometric model described above. The R-squared value is 0.99, which allows us to establish that the model used is robust and appropriate, so we proceed to the interpretation of the coefficients. Similar to the results found by (Nketiah-Amponsah 2009) in Ghana, (Kouton 2018) in Côte d'Ivoire and (Suwandaru et al. 2021) in Indonesia, this study shows that public spending on education has no statistically significant effect on economic growth in Honduras. In this research, government expenditure on education as a percentage of GDP (EDUGDP) obtained a coefficient of 0.1642531 (*p*-value = 0.075), while government expenditure on education as a percentage of government expenditure (EDUEXP) reflects a coefficient of 0.0061939 (*p*-value = 0.550). These results show that investment in education by the government hardly contributes to the increase in productivity and income of the population.

**Table 2.** Empirical results.

| Model | IV/2SLS | |
|---|---|---|
| **Independent Variables** | **Dependent Variable GDPPC** | |
| | Coefficient | Standard Errors |
| RGDPPC | 0.0005761 ** | 0.0002825 |
| GDP | 0.9701458 ** | 0.0057327 |
| INFLA | −0.0102688 ** | 0.003225 |
| UNEMP | 0.0535181 ** | 0.0217097 |
| GFCF | 0.0067179 | 0.0059983 |
| TRADE | 0.0008738 | 0.0028078 |
| FDEPTH | −0.0154494 ** | 0.0040207 |
| INEQUA | −0.0110072 | 0.0083764 |
| EDUGDP | 0.1642531 | 0.0922944 |
| EDUEXP | 0.0061939 | 0.0103679 |
| SCHPRE | 0.0154617 | 0.0110692 |
| SCHPRI | −0.0128421 | 0.0084149 |
| SCHSEC | 0.0146747 ** | 0.0063967 |
| SCHTER | −0.020979 | 0.0184172 |
| STUNT | 0.0583849 | 0.04376 |
| MORTAL | 0.0031108 | 0.0032503 |
| SURVIF | −0.5032859 | 0.3299224 |
| SURVIM | 0.5362868 | 0.3399412 |
| Constant | −3.015047 | 5.17334 |
| **Instrument** URBAN | √ | |
| **Observations** | 31 | |
| **R-squared** | 0.99 | |

** $p < 0.05$.

Of the control variables analyzed in this study, the following are correlated with economic growth, GDP per capita (RGDPPC), unemployment (UNEMP), inflation (INFLA) and financial depth (FDEPTH), with coefficients of 0.0005761 (*p*-value = 0.041), 0.0535181 (*p*-value = 0.014), −0.0102688 (*p*-value = 0.001) and −0.0154494 (*p*-value = 0.000), respectively, although the last two show a negative effect on this. The increase in per capita income is an indispensable condition to achieve economic growth (Elistia and Syahzuni 2018), so a positive correlation is essential. In Honduras, fortunately, this premise is met. In relation to unemployment, its evident statistical link, although positive, is distinguished. Okun's law indicates that this relationship must be inverse or negative to achieve economic growth (Hjazeen et al. 2021), so unemployment is retarding economic development; this

result is also found in recent research by (Ojima 2019) in Nigeria. For its part, inflation is another variable that is negatively affecting economic growth; this finding is consistent with the results of (Álvarez 2016), who managed to evidence the negative relationship of inflation on economic growth in Honduras, having an adverse impact on this. Likewise, for economic growth to occur, it is absolutely necessary that financial depth is closely related and positive (Mérő 2004). This means that a one-percentage-point increase in financial depth would decrease the rate of GDP per capita by 0.00015%.

As for gross fixed capital formation (GFCF), it had no significant influence on economic growth during the study period. However, the coefficient has a value of 0.0067179 ($p$-value = 0.263), which indicates that the effect is positive but very weak; this result coincides with the findings of (Gibescu 2010) in Hungary. For its part, trade openness (TRADE) could encourage economic growth (Frankel and Romer 1999). In the case of Honduras, this variable has no statistically significant relationship on economic development; the coefficient is 0.0008738 ($p$-value = 0.756), showing an inconsequential correlation. According to (Huchet et al. 2018), for countries to experience significant economic growth rates, they must export a variety of and quality products. Similar results were found by (Musila and Yiheyis 2015) in Kenya.

The effect of inequality (INEQUA) from the point of view of income will depend on the country and the region; its influence is negative on economic growth in low and lower middle income countries (Castelló-Climent 2010). This statement is clearly distinguished in Honduras; inequality has a behavior that impedes economic growth. This has no correlation, and the coefficient −0.0110072 ($p$-value = 0.189) implies that the increase in one percentage point of inequality would reduce the rate of GDP per capita by 0.00011%. This revelation is consistent with what was found by (Gründler and Scheuermeyer 2018), who shows that this relationship is insignificant in developing countries. Therefore, this variable is also negatively affecting economic growth in Honduras, probably due to, among other things, the low development of the financial system and low human capital formation (Topuz 2022).

Concerning the variables related to human capital, the rates of gross enrollment in education (preprimary, primary, secondary and tertiary), with the exception of gross enrollment in secondary education, have no significant correlation with economic growth in Honduras. School enrollment, secondary (SCHSEC) has a coefficient of 0.0146747 ($p$-value = 0.022) and is significantly correlated with economic growth; a one-percentage-point increase in this would represent an increase of 0.00015% in the rate of GDP per capita. The same relationship was found by (Pegkas 2014) in Greece and (Adejumo et al. 2021) in Nigeria.

Gross enrollment rates in preprimary, primary and tertiary education have no statistically significant relationship with economic growth in the study period. Surprisingly, these last two have a negative relationship; they not only do not contribute to economic growth but also suggest that educational policy has clearly been inefficient, as well as the public spending directed by the state to these educational levels. The coefficient of school enrollment, preprimary (SCHPRE) is 0.0154617 ($p$-value = 0.162), which indicates that there is no relationship between this variable and economic growth, contradicting most of the empirical evidence. It is important to note that an increase in preschool enrollment would produce economic benefits in adult income (Magnuson and Duncan 2016).

On the other hand, school enrollment, primary (SCHPRI) yields a coefficient of −0.0128421 ($p$-value = 0.162), coinciding with the findings of (Abbas 2001) in his study conducted in Pakistan and Sri Lanka, where he also evidences negative values and a non-significant relationship between these variables. One reason that could cause this is, due to the high levels of poverty, parents would be sending their children to work instead of sending them to school; in addition, in the recent past, irregular migration has displaced a significant number of children of primary school age.

Regarding school enrollment, tertiary (SCHTER), it shows a coefficient of −0.020979 ($p$-value = 0.255), suggesting that there is no correlation and that its effect is negative on

economic growth. This finding is in line with research by (Pereira and St. Aubyn 2004) in Portugal and (Gümüş and Kayhan 2012) in Turkey. Some considerations regarding this finding would be the effect that the Academic Attitude Test (AAT), conducted by the public university as a filter, has had on the poorest segment of the population. This mainly affects students who have graduated from the public secondary education system. In contrast, the demand for places is greater than the supply, mainly in health and engineering careers. Likewise, although there is a notorious incursion of private universities that have made it possible to increase coverage, there is probably still a considerable gap to be covered.

Like Yaya et al. (2020), who showed that prevalence of stunting, height for age (STUNT) is not associated with economic growth in several sub-Saharan African countries, there is no significant relationship between these variables in Honduras either, having a coefficient of 0.0583849 (*p*-value = 0.182). Chronic child malnutrition is a key aspect that could affect child mortality and morbidity, which has broad repercussions on their cognitive development and academic and professional performance, reducing the chances of social advancement (Cusick and Georgieff 2016). The absence of quality medical services and sanitary infrastructure, as well as the poor conditions of energy and water services, coupled with the high cost of living, would be preventing significant economic growth.

The mortality rate, under 5 (MORTAL) is also uncorrelated with the increase in GDP per capita rate, having a coefficient of 0.0031108 (*p*-value = 0.339). Income is an important determinant of child survival (O'Hare et al. 2013). Most scholars who investigated the relationship between these variables reported a negative and inverse relationship, i.e., as income increases, mortality rate decreases. People with more income are healthier; a higher level of income is strongly related to the general health status of the population (Hague et al. 2008). This condition is not registered in Honduras.

After the IV technique (2SLS) tests in Stata, the econometric model takes the following form:

$$
\begin{aligned}
\text{GDPPC} = \ & -3.015047 + 0.0005761\text{RGDPPC} + 0.9701458\text{GDP} \\
& - 0.0102688\text{INFLA} + 0.0535181\text{UNEMP} \\
& + 0.0067179\text{GFCF} + 0.0008738\text{TRADE} \\
& - 0.0154494\text{FDEPTH} - 0.0110072\text{INEQUA} \\
& + 0.1642531\text{EDUGDP} + 0.0061939\text{EDUEXP} \\
& + 0.0154617\text{SCHPRE} - 0.0128421\text{SCHPRI} \\
& + 0.0146747\text{SCHSEC} - 0.020979\text{SCHTER} \\
& + 0.0583849\text{STUNT} + 0.0031108\text{MORTAL} \\
& - 0.5032859\text{SURVIF} + 0.5362868\text{SURVIM}
\end{aligned}
$$

## 5. Discussion

The constitution of Honduras transfers to the government the power to direct and organize education at the basic and intermediate levels of the formal education system, as well as its financing. As for higher education, the National Autonomous University of Honduras is responsible for its management. It is then the function of the state to allocate public spending on education under the precept of responding to the current interests and needs of society, in addition to promoting the process of economic and social development of the country. This study is an attempt to find the relationship between public spending on education, human capital and economic growth. The results obtained show that public spending on education and human capital are inconsequential in the economic development of Honduras. A similar conclusion was reached by (Orozco and Valdivia 2017).

The last educational reform carried out in Honduras was in 2012. However, this does not refer to maintaining a balance in funding or prioritizing investments in innovation, research, construction or technology. A simple analysis shows that current spending represented 94% of public spending on education in 2020 (Tribunal Superior de Cuentas 2021). This essentially refers to the payment of salaries and wages, which leaves limited options for investment in equipment and infrastructure. The same conclusion is reached by

(ICEFI 2017), who identify a significant gap in equipment and infrastructure investments. Therefore, it can be assumed that public spending on education has, as its greatest benefit, the hiring of employees in the education sector, this benefit being greater than the proper contribution that education should have on the socioeconomic development of the country.

The current structure of public spending on education in Honduras favors primary education and the payment of salaries and wages. This is to the detriment of preschool and secondary education. The low capital expenditure planning triggers severe infrastructure problems due to the notable deterioration of educational centers, while reducing the quality of education. These assessments coincide with a document prepared by (World Bank 2015). Regarding human capital, evidence indicates that this is a major constraint for economic growth in Honduras. Low levels of education prevail, deterring business investment (USAID 2021).

## 6. Conclusions

The results obtained in this research could be interpreted as an approximation to the findings previously found by Kiran (2014), whose results suggest that the evidence is inconclusive in the case of Honduras regarding the relationship between public expenditure for education and economic growth. The present study determines that there is no correlation between both variables; however, there seems to be a weak link when public expenditure for education is decomposed, particularly with regard to government expenditure on education as a percentage of GDP, given that increasing this variable by one point represents an increase of 0.1642531 in the rate of GDP per capita. Given this discouraging scenario, it is possible to assume that education policy during the research period has not responded to the needs and expectations of the population. The management of public spending on education, despite being a tool available to the government to generate welfare and prosperity, has been unfocused and has been wasted; its execution reflects signs of negligence, since it has not been accompanied by a significant increase in income or productivity of citizens.

The results found suggest that human capital is not contributing to economic growth in Honduras; enrollment rates in primary and tertiary education are having a negative influence, which is very surprising because it confirms that the accumulation of human capital is not being fully developed. The government needs to conduct a thorough analysis of the management of education and health systems as the main means that states have to channel resources to raise the productivity of its citizens. A country with low labor productivity will have high unemployment and poverty rates, in addition to low economic growth (Vijayakumar 2013).

A country's economic growth rates depend on and are systematically related to a set of quantifiable explanatory variables (Barro and Lee 1994). There are several aspects that have been addressed and explained in this study that are affecting economic growth in Honduras, in addition to public spending on education and human capital. Variables such as inflation, financial depth, inequality, unemployment, trade openness and gross fixed capital formation should be examined by decision makers, since all these elements are considered key by the literature to promote the social and economic development of a country, and in this case, they would not be promoting it. These findings only come to expose what was already suspected, so that the government and the population in general have enormous challenges to overcome in an increasingly complex global context.

There are enormous challenges arising from the absence of reliable information and data in public institutions. In addition, there is basically no formal research based on empirical evidence that deals exclusively with Honduras in this area. It is hoped that this study can become a reference for new researchers. The consideration of these issues in the analysis of the relationship between public education expenditure, human capital and economic growth in Honduras has brought about new revelations. New lines of research related to the effect of current and capital spending on education on human capital accumulation and economic growth should be explored. Similarly, following the literature

and in view of the findings of this study, it is necessary to analyze the role of education as a determinant of economic growth in Honduras.

**Author Contributions:** Conceptualization, R.V. and J.J.P.; methodology, J.J.P.; software, R.V.; validation, R.V. and J.J.P.; formal analysis, R.V.; investigation, R.V.; resources, R.V.; data curation, R.V.; writing—original draft preparation, R.V.; writing—review and editing, R.V. and J.J.P.; visualization, R.V.; supervision, J.J.P. All authors have read and agreed to the published version of the manuscript.

**Funding:** This research received no external funding.

**Informed Consent Statement:** Not applicable.

**Data Availability Statement:** Not applicable.

**Acknowledgments:** We thank the Universidad Tecnologica Centroamericana and its team for their support.

**Conflicts of Interest:** The authors declare no conflict of interest.

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
