# Peer review of "Empirical Analysis on Public Expenditure for Education, Human Capital and Economic Growth: Evidence from Honduras"

_economies, doi:10.3390/economies10100241_

Round 1
Reviewer 1 Report
I consider the paper to be well written. It is quite extensive. These is a rather basic type of research report.
The theoretical part is well prepared, it contains a large number of resources.
Regarding the method, I suggest that the authors better justify the choice of method.
A much more detailed analysis needs to be done. Table 2 is completely inadequate. Authors should be more ambitious and ask more research questions. Research does not bring much new.
Why Honduras and what can be learned from this case study should be stated??
There is no discussion at all. Without discussion, it is not possible to publish the paper at all.
Authors should be given the opportunity to extensively edit the paper, especially the research part, and they must supplement the discussion.
Limitations and directions for further research should be added to the conclusion.
Reviewer 2 Report
Dear authors,
The article deals with an interesting and important issue. However, I have several comments on it, which could improve the quality and readability of your paper.
You use very long sentences. For example, lines 35 - 40 is only one sentence, also lines 41 - 47 or 95 - 100. It would be better to divide them into more sentences because now it is hard to follow.
In lines 178 - 186, you mentioned the variables used in the model and that they were used in other studies. Please, add the references for such studies using these variables. Also, I don't understand the sentence in lines 184 - 194. Please, reformulate it.
Please add the paragraph about the structure of the following sections of the paper at the end of the introduction section.
I miss the references in the introduction section. From where do the numbers come from?
The in-text citations are made incorrectly. For example, line 79: (Okerekeoti 2022) examined the effect of public expenditure... Correct is Okerekeoti (2022) examined the effect of public expenditure...There are more such citations; please correct them.
Please do not use abbreviations without an explanation, for example, ARDL (line 89). Not all your readers know what it is.
I did not understand what instrumental variable(s) did you used. It is probably mentioned in the text, but it is very weakly emphasized for me. You probably used a 2-stage model; what steps did you use? Logit or probit model? Please describe it in more detail. Accorging to the text in lines 202 - 213 I have an impression that you used only the econometrical model, without the instrumental variable method. Did you interpret the values of the coefficients from the first step or the second? Please clarify.
I also miss the test on the weak instrument. What is the correlation between the instrument and the dependent variable?
In lines 215 - 219, you mention a high correlation between the variables. However, I do not consider the correlation with a coefficient of 0.0005761 to be a high correlation.
There is no discussion section in the paper. In the literature review, you mentioned several studies aimed at the same or similar issue. I think that a comparison of your results with similar studies made in lines 245 - 311 should be a part of the discussion section. What are the weaknesses of your study, and how should the results be used in practice? Where do you see the possible continuation of the study?
Round 2
Reviewer 1 Report
Thanks for all revisions and your effort to do it according to my reccommendations. Good job.
Reviewer 2 Report
Dear Authors,
Thank you for your effort. I consider the article in its current form publishable.